# Microbiological and Immunological Markers in Milk and Infant Feces for Common Gastrointestinal Disorders: A Pilot Study

**DOI:** 10.3390/nu12030634

**Published:** 2020-02-27

**Authors:** Marina Aparicio, Claudio Alba, Juan Miguel Rodríguez, Leonides Fernández

**Affiliations:** 1Department of Nutrition and Food Science, Complutense University of Madrid, 28040 Madrid, Spain; marinaap@ucm.es (M.A.); jmrodrig@vet.ucm.es (J.M.R.); 2Department of Galenic Pharmacy and Food Technology, Complutense University of Madrid, 28040 Madrid, Spain; claudioalbarubio@gmail.com; 3EAP Valle de la Oliva, Majadahonda, 28220 Madrid, Spain; enriqueta.roman@salud.madrid.org

**Keywords:** human milk, infant feces, microbiome, immunology, intestinal diseases, colic, non-IgE-mediated allergy, proctocolitis, pilot study

## Abstract

The objective of this pilot study was to assess the fecal microbiome and different immunological parameters in infant feces and maternal milk from mother–infant pairs in which the infants were suffering from different gastrointestinal disorders (colic, non-IgE-mediated cow milk protein allergy (CMPA), and proctocolitis). A cohort of 30 mother–infant pairs, in which the infants were diagnosed with these gastrointestinal disorders or included as healthy controls, were recruited. Bacterial composition of infant feces and breast milk was determined by metataxonomic sequencing. Immunological compounds were quantified using multiplexed immunoassays. A higher abundance of *Eggerthellaceae*, *Lachnospiraceae* and *Peptostreptococcaceae*, and lower abundance of *Bifidobacterium* and higher abundance of *Rothia* were registered in fecal samples from the CMPA group. *Eggerthellaceae* was also significantly more abundant in milk samples of the CMPA group. There were no differences in the concentration of immunological compounds in infant fecal samples between the four groups. In contrast, differences were found in the concentration and/or frequency of compounds related to acquired immunity and granulocyte colony stimulating factor (GCSF) in breast milk samples. In conclusion, a few microbial signatures in feces may explain part of the difference between CMPA and other infants. In addition, some milk immunological signatures have been uncovered among the different conditions addressed in this pilot study.

## 1. Introduction

Infant gastrointestinal disorders are defined as different combinations of chronic or recurrent gastrointestinal signs and symptoms that usually do not correspond to structural or biochemical disorders [1]. Gastrointestinal disorders occur in a high proportion of the infants and are a common source of infant and parental discomfort as they lead to reiterated visits to the pediatrician, at either the primary care or hospital level, and frequent and unsuccessful changes in infant feeding practices [2]. The term includes diverse disorders, such as colic, non-IgE-mediated allergies and proctocolitis.

Infantile colic is defined as infant paroxysms of inconsolable crying and/or fussing for more than three hours per day, more than three days per week [3]. This condition involves several gastrointestinal factors, such as intestinal immaturity, hypermotility secondary to a presumed autonomic imbalance and alterations in the fecal microbiome [4]. Moreover, it increases the probability of suffering from psychosocial distress and depression, negatively affecting the maternal–child bond [5,6]. Non-IgE mediated food allergy has also emerged as a frequent disease in recent years, involving cellular immunity and mainly affecting the digestive tract [7]. This disorder is characterized by subacute and/or chronic non-specific symptoms that appear gradually (from one hour to several days after contact with the antigen), and include abdominal colicky pain, inappetence, pathological gastroesophageal reflux, blood in stools, diarrhea, constipation and, less frequently, enteropathy with poor weight gain [4]. Cow’s milk protein allergy (CMPA) is the most prevalent type of food allergy in infancy, and its incidence during first year of life has been estimated to be 2% to 7.5% [8]. Proctocolitis accounts for up to 64% of rectal bleeding in the infant, and CMPA or lactose intolerance plays a major role in its development [9]. It is one of the most frequent food allergies but these children do not show signs of systemic involvement (vomiting, diarrhea, fever or growth disorders), which separates this disorder from other manifestations of allergy [4]. Unfortunately, the pathogenesis of the conditions cited above is still poorly understood [10] and they usually share some symptoms; consequently, the differential diagnosis remains a challenge.

Alterations in both the innate immune system and the intestinal microbiota, which play an important role in the development of intestinal tolerance, seem to be involved in these conditions [11,12,13], but their potential link is still far from elucidated. The mature gastrointestinal tract is typically an effective barrier to prevent intact ingested food antigens from stimulating the immune system. However, this barrier is immature in newborns and these intact antigens are able to permeate the intestinal wall and induce an immune response [14]. The intestinal microbiota plays a crucial role in the development of the immune response of the intestinal mucosa and alterations in the early gut microbiota may precede the development of allergies [13] as well as the acquisition of oral tolerance [15]. Human milk also plays a key role in shaping the gut microbiota of infants because it provides a continuous source of commensal microorganisms and some compounds that act as prebiotics [16,17]. Therefore, gut microbiota could be both a relevant target for innovative therapeutic strategies in children with gastrointestinal disorders and a source of simple and non-invasive markers for these conditions.

In this context, the objective of this work was, first, to assess the microbiome and a wide spectrum of immunological parameters in maternal milk and infant feces from groups of mother–infant pairs in which the infants were suffering from any of the different gastrointestinal disorders described above. Secondly, to elucidate if there are immunological or microbiological condition-specific signatures.

## 2. Materials and Methods 

### 2.1. Participants

A cohort of 30 mother–infant pairs, in which the respective infants were diagnosed with colic (*n* = 12), non-IgE mediated CMPA (*n* = 5), proctocolitis (*n* = 5), or included as healthy controls (*n* = 8), were recruited in this pilot study (Table 1). In accordance with the Declaration of Helsinki, all volunteers, parents or legal tutors gave written informed consent to the protocol, which had been approved (protocol 10/2015) by the Norwest Local Ethical Committee of Clinical Research of Servicio Madrileño de Salud (Madrid, Spain). Case definitions, inclusion and exclusion criteria, and the main characteristics of the recruited infants are shown in Table 1.

### 2.2. Collection of the Samples

Milk and infant fecal samples were collected as previously described [18], and stored at −20 °C in the Primary Health centers. Samples were shipped in temperature-controlled containers to the Complutense University of Madrid for the subsequent analysis.

### 2.3. DNA Extraction from the Samples

Approximately 1 g of fecal or milk sample from each infant or mother, respectively, was used for DNA extraction following the methods described previously [18]. Extracted DNA was eluted in 22 μL of nuclease-free water and stored at −20 °C until further analysis. Purity and concentration of each extracted DNA sample was estimated using a NanoDrop 1000 spectrophotometer (NanoDrop Technologies, Inc., Rockland, USA). Negative controls were added (blanks) during the extraction.

### 2.4. PCR Amplification and Sequencing

The analysis of the microbiome was carried out using the technique of massive sequencing of the 16S rDNA gene through the MiSeq system of Illumina (Parque Científico de Madrid), with the universal primers S-D-Bact-0341-b-S-17 (CCTACGGGNGGCWGCAG) and S-D-Bact-129 0785-a-A-21 (GACTACHVGGGTATCTAATCC) directed to the V3-V4 hypervariable regions of the 16S rDNA [19]. In the second PCR-reactions, forward and reverse sequences were separated by barcodes appended to 3’ and 5’ terminal ends of the PCR amplicons. The concentration of DNA for each sample was measured by the 2100 Bioanalyzer system (Agilent, Santa Clara, CA, USA). PCR products were pooled at around equimolar DNA concentrations. The band that had the correct size after running on the agarose gel, was excised and purified by using a QIAEX II Gel Extraction Kit (Qiagen) and then quantified with PicoGreen (BMG Labtech, Jena, Germany). The pooled, purified and barcoded DNA amplicons were sequenced by using the Illumina MiSeq pair-end protocol (Illumina Inc., San Diego, CA, USA) following the manufacturer’s protocols at the facilities of Parque Científico de Madrid (Tres Cantos, Spain). Negative extraction control was used in PCR reactions and their products were not sequenced after not showing visible bands in the electrophoresis (1% agarose gel). The Illumina reads or amplified fragments (forward and reverse) were combined into single reads by using SeqPrep (https://github.com/jstjohn/SeqPrep) with a limit of 0.5 mismatched bases. After quality control, the reads were assembled and the resulting sequences were processed by using QIIME package version 1.9.1 [20,21] and classified taxonomically into operational taxonomic units (OTUs) by comparison with the SILVA SSU database (version 132) [22] using a Naïve-Bayes classification method.

### 2.5. Immunological Assays

Fecal samples (approximately 0.1 mL) were suspended in PBS (1:10 w/v) and homogenized. Supernatants collected after centrifugation (14,000 × *g*, 15 min, 4 °C) were used for determination of cytokines, chemokines and growth factors in duplicate by the BioPlex system (BioRad), using the Human Cytokine, Chemokine and Growth Factor Assays kit as described previously [23]. These include: interleukins (IL) 1β, 2, 4, 5, 6, 7, 8, 10, 12(p70), 13 and 17, interferon-gamma (IFNγ), granulocyte colony stimulating factor (GCSF), granulocyte-macrophage colony stimulating factor (GMCSF), monocyte chemoattractant protein-1 (MCP1), macrophage inflammatory protein-1β (MIP1β) and tumor necrosis factor-alpha (TNFα). Standard curves were performed for each analyte. Centrifugation (800 × *g*, 15 min, 4 °C) was used to remove the fat layer and suspended material from milk samples. The intermediate aqueous layer was collected for determination of the same immunological compounds.

### 2.6. Statistical and Bioinformatic Analysis

Normally distributed quantitative data were expressed as the mean and 95% confidence interval (CI) of the mean. When not normally distributed, the data were presented as the median and interquartile range (IQR). Kruskal–Wallis tests were used to evaluate the differences in the median concentration of the immunological compounds and in the median relative abundances of dominant taxa, followed by Dunn’s tests to identify which levels of the independent variable differ from each other level. Fisher’s exact tests or the Freeman–Halton extension of the Fisher exact probability test for a 2 × 4 contingency table were performed to compare the frequency of detection of different bacterial genera and immunological compounds. 

LefSe analysis (LDA effect size) was used to identify differences in taxonomy between the study groups [21]. The richness and diversity of the milk and fecal microbiota were determined by calculating the Shannon diversity index, which takes into account the number and evenness of the bacterial species. Microbiota community differences between samples (beta diversity) were tested by permutational multivariate analysis of variance (PERMANOVA) comparison of unweighted UNIFRAC distance matrices, with 999 permutations. Spearman rank correlation was performed to evaluate possible correlations between different factors of study (R package: *corrplot*). Agglomerative or cluster-merging hierarchical clustering was performed by using the Euclidean distance and complete hclust_method methods (R package: *heatmaply*) to analyze the binary matrix of detection of the immune factors evaluated in the study. Afterwards, a heatmap was constructed including the detection of the immune factors in the different sample types and the sample description, as well as colored bars vector to classify the samples in the different study groups. All analyses were performed with the R software version 3.3.2 (R-project, http://www.r-project.org).

## 3. Results

### 3.1. Metataxonomic Profiling of the Fecal Samples

The most frequently detected and abundant genus in the fecal samples was *Bifidobacterium*, which was present in nearly all the samples ranging from a median (IQR) relative abundance of 64.24% (61.91%–75.05%) in the control group to 23.40% (5.27%–51.23%) in the CMPA group (Table 2 and Figure 1a). *Escherichia-Shigella*, *Streptococcus* and *Lactobacillus* were also detected in >50% of the samples (Table 2 and Figure 1a). No differences were observed with regard to the frequency of detection or concentration of the most abundant bacterial phyla between groups (results not shown). However, at the family level, the concentration of *Eggerthellaceae*, *Lachnospiraceae* and *Peptostreptococcaceae* was higher in fecal CMPA samples than in those from other groups (Figure 1b). At the genus level, when comparing groups by pairs, lower abundance of *Bifidobacterium* and higher abundance of *Rothia* was the main feature of the CMPA group in comparison with the other three groups. Moreover, *Erysipelatoclostridium* abundance was higher in controls and *Intestinibacter* was higher in the proctocolitis group followed by the CMPA one (Figure 1c).

The median number of the observed species was significantly higher in fecal samples of the CMPA group when compared with samples of the control group (Figure 2a; *p* < 0.001). Differences were not observed in the bacterial diversity calculated as the Shannon Index (Figure 2b). The analysis of the beta diversity in samples of the three disease groups together did not reveal distinct microbial profiles when compared to samples of the control group (Figure 2c,d).

### 3.2. Metataxonomic Profiling of the Milk Samples

The most abundant bacterial genera in milk samples were *Streptococcus* and *Staphylococcus*, followed by genera of the family *Enterobacteriaceae* (Figure 3a and Table 3). *Acinetobacter*, *Corynebacterium*, *Lactobacillus*, and *Gemella* were also present but at lower concentration. Similar to what has been described above for the fecal samples, the most abundant genera were detected in all the study groups and their detection frequency and concentrations did not differ significantly between the study groups (Table 3). However, in paired comparisons between groups, the relative abundance of the family *Eggerthellaceae* was significantly higher in CMPA than in the control and colic groups (Figure 3b). In addition, *Prolixibacteraceae* was higher in the proctocolitis group than in the CMPA group (Figure 3b).

Regarding the alpha diversity of milk samples, no differences were obtained when comparing the bacterial richness and the Shannon index values due to the high interindividual variability of the samples (Figure 4a,b, respectively). The beta diversity analysis did not reveal characteristic microbial profiles for the control group or for the three disease groups together (Figure 4c,d).

### 3.3. Concentration and Frequency of Detection of Immune Factors in the Infant Fecal Samples

All the immune compounds were detected at least in one of the infant fecal samples analyzed (Table 4). IL12, IL2, IL17 and IL5 were present in all the fecal samples, and IL1β, IL6, IL8, IL10, GCSF, GMCSF, MIP1β and TNFα were detected in more than 50% of them. GCSF was found at a lower frequency in the CMPA and proctocolitis groups (60% and 75%, respectively) than in the control and colic groups (100%) (*p* = 0.036) (Table 4).

There was a large interindividual variability in the concentration of the immunological compounds of infant fecal samples. The median (IQR) value of IL10 content in fecal samples from the proctocolitis, CMPA, and control groups were 8.00 (5.80–11.50) ng/L, 937.42 (472.71–1105.06) ng/L, and 34.60 (27.10–42.209) ng/L, respectively, but the differences did not reach the statistical significance (Table 4). A similar observation could be made about the median (IQR) value of IL7 concentration in feces of infants with colic and controls that were 8.60 (8.60–35.90) ng/L and 86.30 (74.00–100.37) ng/L, respectively (Table 4).

### 3.4. Concentration and Frequency of Detection of Immune Factors in Milk Samples

Globally, all the immune compounds were also present at least in one of the milk samples analyzed (Table 5). MIP1β was the only factor present in all samples, and IL1β, IL6, IL8, IL12, IL13 and MCP1 were found in more than the 50% of the samples. When all groups were compared, statistically significant differences were obtained in the concentration and/or frequency of all the compounds related to acquired immunity (IL2, IL4, IL10, IL13, IL17) and GCSF (Table 5). IL2 was only detected in milk samples of the CMPA group, and IL10 and IL17 concentrations were also higher in this group (median (IQR) concentrations of 24.00 (13.51–29.46) ng/L and 20.19 (14.15–26.23) ng/L, respectively). On the other hand, IL13 concentration was lower in mother’s milk of the infants with colics when compared with the rest of the groups (Table 5).

### 3.5. Relationship between the Immune Profile of Milk and Infant Fecal Samples

Overall, the frequency of detection of all immunological compounds, and in particular those of IL6, IL12, IL2, IL10, IL17, IL5, GCSF and GMCSF, was higher in the infant feces than in milk samples, while the opposite occurred for IL4, IL13, IL8, MCP1, and MIP1β (Appendix A). Most of the immunological compounds were detected in both types of samples roughly at the same proportion in feces and milk samples between individual study groups with the exception of IL2, IL5, and IL17 that were absent from most milk samples (Table 6). In addition, in the colic group the frequencies of detection in feces for IL6, IL12, IL10, and GCSF were significantly higher than in milk (*p* < 0.005), while the opposite was observed for IL8 and MCP1 (*p* = 0.037 and *p* = 0.012, respectively). (Table 6).

There was no clear clustering according to the type of sample (feces or milk) nor to the clinical condition of the immunological profile of fecal and milk samples, according to the presence or absence of the immunological compounds (Appendix A).

## 4. Discussion

In this pilot study, we evaluated the microbiological and immunological profiles of samples of feces and milk obtained from mother–infant pairs in which the respective breastfed infants were suffering from different gastrointestinal disorders. Intestinal microbiota plays a key role in the host metabolism and in the maturation and education of the immune system [24]. Several studies have assessed the fecal microbiota in infants with different gastrointestinal disorders (colic, food allergy and/or food hypersensitivity), and all of them postulated an association between these conditions and a gut dysbiosis state [11,12,13,15]. 

*Bifidobacterium* is considered as the dominant bacterial genera in the infant gut microbiome of breastfed infants [25,26]; in our study, the abundance of *Bifidobacterium* DNA in the feces of CMPA infants was lower than that observed in the other groups. A reduced presence of bifidobacteria in fecal samples from infants suffering CMPA has already been reported [11,27]. This finding might be related to the type of delivery. Most of the infants in CMPA group were born by Cesarean section, which has been related to lower *Bifidobacterium* content due, at least in part, to delayed initiation of breastfeeding [28]. Moreover, in this study fecal samples of CMPA infants were also characterized by a higher abundance of the genus *Rothia* and the families *Lachnospiraceae*, *Peptostreptococcaceae* and *Eggerthellaceae*. *Lachnospiraceae* and *Peptostreptococcaceae* have been previously reported to be enriched in the gut microbiota of CMPA infants, but the significance of their presence remains unclear [15,26,29,30,31,32,33]. Remarkably, *Eggerthellaceae* was more abundant in the milk of mothers of CMPA infants. *Eggerthellaceae* is an emerging bacterial family that has been associated to gastrointestinal and genitourinary pathologies and some members of this family are characterized by their ability to spread from these locations to blood [34,35], a fact that would explain their presence in milk samples. Unfortunately, maternal fecal samples were not available in this study and, therefore, we could not assess potential mother-to-infant transfer of members of this family.

In our study, bacterial diversity was higher among the feces of CMPA infants than among those from healthy controls. This result is in agreement with that of a previous study which reported an increased microbial diversity in children with milk allergy in comparison to healthy ones [15]. In contrast, other authors [13] found a reduced diversity of the early intestinal microbiota of infants with allergic diseases using DNA fingerprinting techniques, which can only detect bacteria with a relative abundance of >1%. Globally, our results showed that infants with CMPA present a peculiar fecal microbial profile, which is different from that of the other groups of infants included in this study (colic, proctocolitis, and healthy). However, no statistically significant differences were found among the later groups despite previous studies had reported the existence of a bacterial dysbiosis and a reduced microbial diversity in the feces of infants suffering from lactational colic [6,36,37,38].

Breast milk contains immunologically active compounds that provide a protective effect for CMPA in children at high risk [25,39], being one of the main drivers in determining the microbiome composition of the infant gut [17,25,40]. Unfortunately, studies addressing the mechanisms enabling human milk to reduce the allergic disease risk are very scarce [41].

The immature gastrointestinal barrier in newborns is ineffective to prevent intact ingested food antigens from stimulating the immune system. These antigens permeate the intestinal wall inducing an immune response [14]. Moreover, bacterial dysbiosis predisposes the neonatal intestine to inflammation [42]. For this reason, a wide range of cytokines, chemokines, growth factors and immunoglobulins were determined in the fecal and milk samples in order to characterize their immunological profiles and their possible relation with bacterial colonization. Our study did not reveal notorious changes in either the frequency of detection or the concentration of the immunological factors in feces, except for a lower frequency of detection of GCSF in fecal samples of infants with CMPA and proctocolitis. This growth-stimulating factor is released when there is a bacterial infection to increase the maturation of neutrophils [43]. In addition, IL7 concentration was lower in samples of infants with colics in comparison to those from the control group. IL7 is a non-hematopoietic cell-derived cytokine with a crucial role in the adaptive immune system and for B and T cell development and may serve as a regulatory factor for intestinal mucosal lymphocytes [44]. Another immunological feature of fecal samples from infants with proctocolitis was a reduced content of IL10. IL10 suppresses both the innate and adaptative responses of the immune system, and limits the inflammatory responses [45]. Thus, reduced fecal IL10 could explain bleeding related to gut inflammation in the Proctocolitis group.

In contrast with the fecal samples, milk samples showed variations among the four study groups in the immunological factors related to acquired immunity. IL2, which plays an important role in balancing the immune response [46], was only detected in CMPA group. Besides, anti-inflammatory IL10 and pro-inflammatory IL17 and GMCSF were also more abundant in milk from CMPA group in comparison with the rest of the groups. IL10 in human milk seems to provide some degree of protection against CMPA although this relation is unclear [25,47]. On the other hand, IL17 is considered to play an important role on the inflammatory process, but its role (and that of IL17-producing cells) during allergic inflammation remains unknown [48]. IL13 abundance was lower in the milk of mothers whose children suffered from lactational colic. This cytokine has been recently recognized for its novel role in allergic and other inflammatory diseases, being released by signals from an inflamed gut epithelium [49]. Recent studies have shown that selected bacterial species and their metabolites (such as short-chain fatty acids) may positively modulate immune tolerance mechanisms [33]. Although some studies have postulated a possible association between allergy and an altered microbiota pattern [11], the gut microbiota of infants suffering non-IgE-mediated CMPA remains uncharacterized [30], partly because of the difficulty in establishing an unambiguous diagnosis [31].

The most important limitation in this pilot study is the low size of the cohort sample, which is mainly related to the in practice difficulties in a proper differential diagnosis among the gastrointestinal disorders included in this study. However, we identified a few microbial signatures in feces that may explain part of the difference between CMPA and other infants. We also detected some milk immunological signatures among the different conditions addressed in this study. Their usefulness as biomarkers should be tested in further studies involving a much larger population.

## 5. Conclusions

This study provides preliminary data on the fecal microbiological and immunological profile of gastrointestinal disorders in infants, including colic, CMPA and proctocolitis, and their relation to the microbiological and immunological profile of maternal milk. Only infants in the CMPA group displayed significant variations in the composition of the fecal microbiome, specifically of the family (*Eggerthellaceae*, *Lachnospiraceae* and *Peptostreptococcaceae*) and genus (*Bifidobacterium, Rothia*) taxonomic levels. In addition, there was a higher content of *Eggerthellaceae* in mother’s milk in infants with CMPA. The immunological profile of milk samples of CMPA group was also distinct with regards to all immunological compounds related to the acquired immunity assayed in this study (IL2, IL4, IL10, IL13, and IL17). Higher concentrations in milk of this set of cytokines compared to other groups could favor excessive inflammation in the gut.

## Figures and Tables

**Figure 1 nutrients-12-00634-f001:**
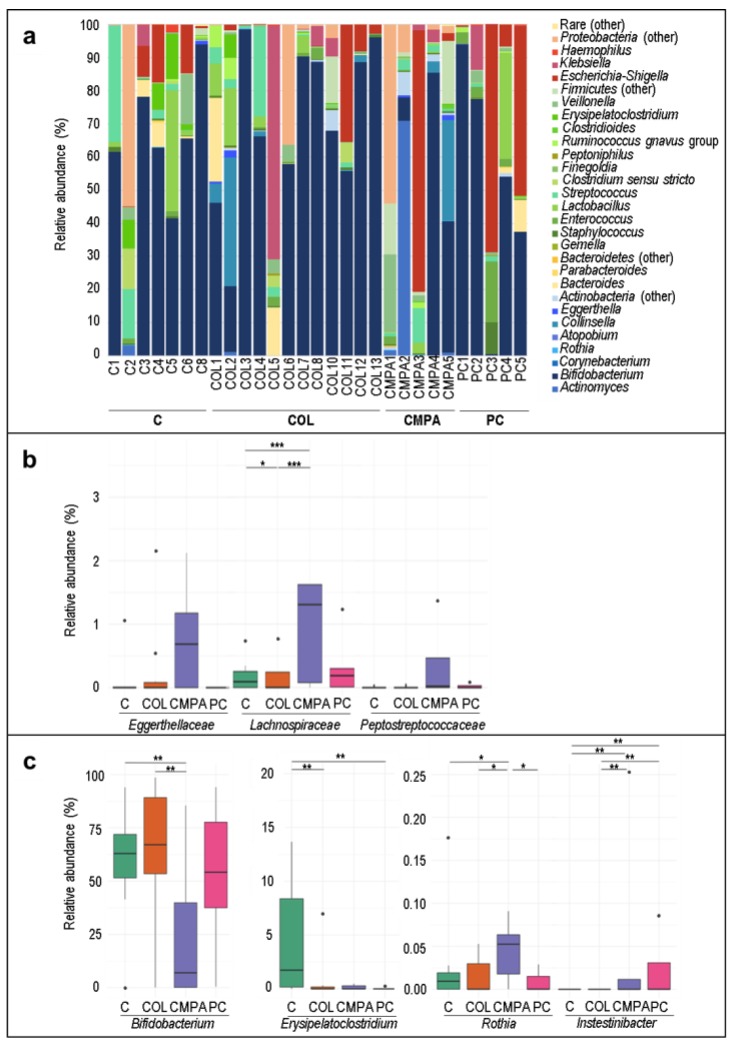
Bacterial composition of infant fecal samples: (**a**) Relative abundance of the major bacterial genera detected in ≥ 15% of the fecal samples; (**b**) Relative abundances of the significantly different bacterial families and (**c**) Relative abundances of the significantly different bacterial genera resulting from LEfSe analysis of the fecal samples (*N* = 29). Statistical differences between the study groups are indicated with an asterisk (*, *p* < 0.10; **, *p* < 0.05; ***, *p* < 0.01; Dunn test). C, Control group (*n* = 7); COL, Colic group (*n* = 12); CMPA, Cow’s milk protein allergy group (*n* = 5); PC, Proctocolitis group (*n* = 5). In these boxplots, the central rectangle represents the interquartile range (IQR), the line and the cross inside the rectangle show the median and the mean, respectively; the whiskers indicate the maximum and minimum values, and the black dots outside the rectangles are suspected outliers (>1.5 × IQR).

**Figure 2 nutrients-12-00634-f002:**
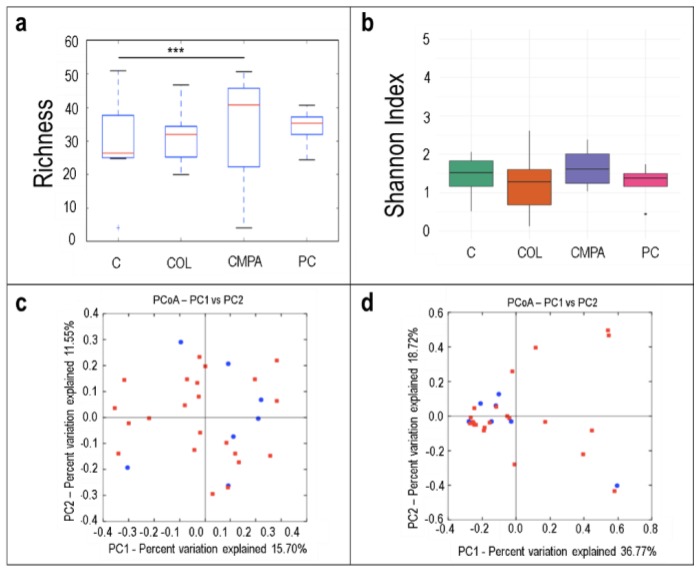
Alpha diversity in the fecal samples (*N* = 29) in the different study groups expressed by: (**a**) the number of the observed species (species richness) and (**b**) the Shannon Index. Beta diversity analysis of the study groups measured as: (**c**) the presence/absence (Binnary Jaccard method; *p* = 0.917) and (**d**) the relative abundance (Bray Curtis method; *p* = 0.839) of the different species quantified in all disease groups (red) and controls (blue). C, Control group (*n* = 7); COL, Colic group (*n* = 12); CMPA, Cow’s milk protein allergy group (*n* = 5); PC, Proctocolitis group (*n* = 5).

**Figure 3 nutrients-12-00634-f003:**
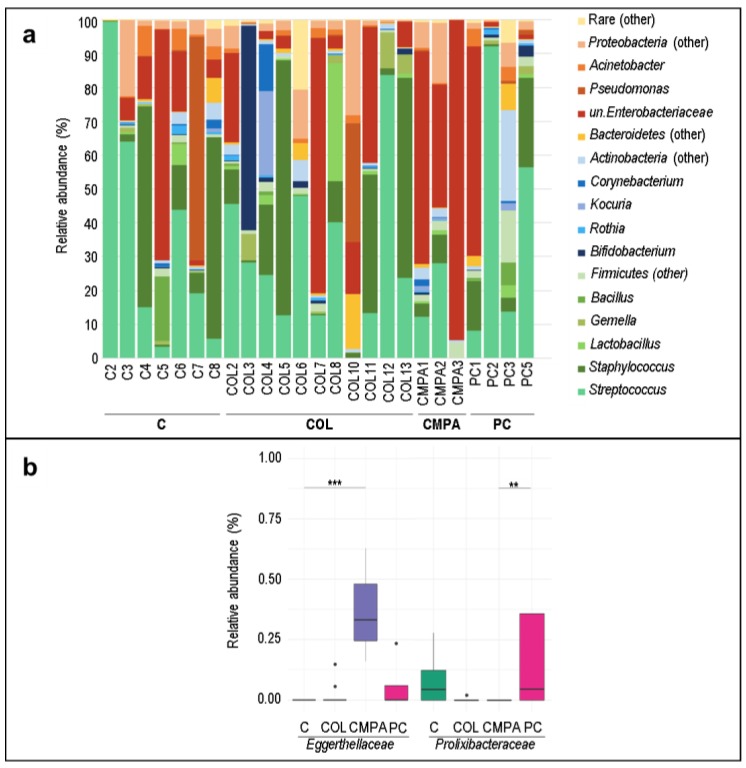
Bacterial composition of milk samples: (**a**) Relative abundance of the major bacterial genera detected in ≥ 15% of the milk samples; (**b**) Relative abundances of the significantly different bacterial families resulting from LEfSe analysis of the milk samples (*N* = 25). Statistical differences between the study groups are indicated with an asterisk (**, *p* < 0.05; ***, *p* < 0.01; Dunn test). C, Control group (*n* = 7); COL, Colic group (*n* = 11); CMPA, Cow’s milk protein allergy group (*n* = 3); PC, Proctocolitis group (*n* = 4). In these boxplots, the central rectangle represents the interquartile range (IQR), the line and the cross inside the rectangle show the median and the mean, respectively; the whiskers indicate the maximum and minimum values, and the black dots outside the rectangles are suspected outliers (>1.5 × IQR).

**Figure 4 nutrients-12-00634-f004:**
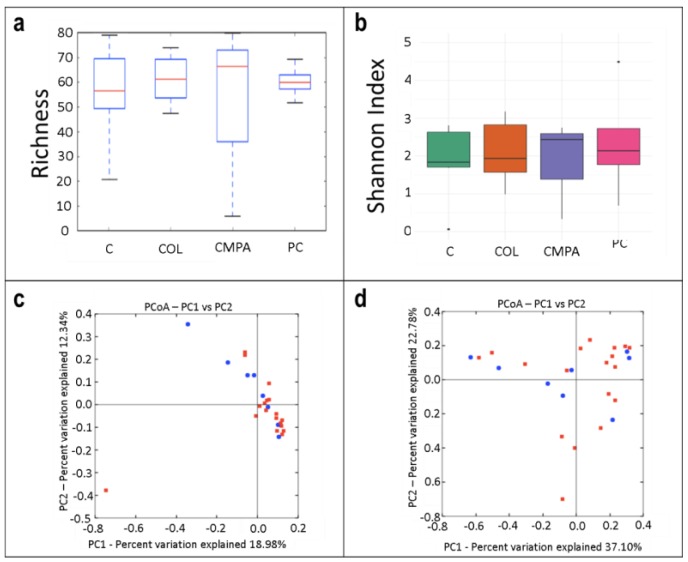
Alpha diversity in the milk samples (*N* = 25) in the different study groups expressed by: (**a**) the number of the observed species (species richness) and (**b**) the Shannon Index. Beta diversity analysis of the study groups measured as: (**c**) the presence/absence (Binnary Jaccard method; *p* = 0.794) and (**d**) the relative abundance (Bray Curtis method; *p* = 0.912) of the different species quantified in all disease groups (red) and controls (blue). C, Control group (*n* = 7); COL, Colic group (*n* = 11); CMPA, Cow’s milk protein allergy group (*n* = 3); PC, Proctocolitis group (*n* = 4).

**Table 1 nutrients-12-00634-t001:** Demographic and clinical characteristics of the four groups of participating infants (*N* = 30), including case definition and inclusion and exclusion criteria.

Characteristic	Control Group	Colics Group	CMPA Group	Proctocolitis Group
*n*	8	12	5	5
Gestational age (wks)	30.38 ± 15.15	28.25 ± 16.85	31.40 ± 14.82	36.00 ± 8.97
Weight (kg)	4.70 ± 1.03	4.30 ± 0.74	4.21 ± 0.72	5.84 ± 1.40
Length (cm)	54.84 ± 3.05	54.29 ± 3.21	54.60 ± 2.58	59.60 ± 5.41
Delivery mode, *n* (%)				
Vaginal	7 (88)	11 (92)	1 (20)	3 (60)
Cesarean	1 (12)	1 (8)	4 (80)	2 (40)
Gender, *n* (%)				
Male	4 (50)	5 (42)	4 (80)	3 (60)
Female	4 (50)	7 (58)	1 (20)	2 (40)
Case definition		Paroxysms of irritability, restlessness or crying at least 3 h/d and at least 3 d/wk, that begin and end without apparent cause in an infant during the first 4 mo with good general condition and good weight gain	Infants with non-IgE-mediated cow’s milk protein allergy	Paroxysms of irritability, restlessness or crying at least 3 h/d and at least 3 d/wk, that begin and end without apparent cause in an infant during the first 4 mo with good general condition and good weight gain
Inclusion criteria	Healthy infants Healthy mother No dietary intervention in the mother	Infants <4 mo Exclusive breastfeeding No dietary interventions in the mother	Infants <6 mo Artificial or mixed breastfeeding (50% of calories contributed by formula) Suggestive symptoms (regurgitation, vomiting, refusal of food, diarrhea, constipation, perianal erythema…) Improvement with cow’s milk protein exclusion diet	Infants <6 mo with clinical symptoms compatible with colitis Exclusive breastfeeding No dietary intervention in the mother Negative stool microbiological study
Exclusion criteria	Any diagnosed or suspected pathology	Infants >4 mo Formula-fed infants Presence of signs or symptoms of other pathologies	Infants >6 mo IgE-mediated CMPA Severe atopic dermatitis probably related to cow’s milk proteins (as a single manifestation) Enterocolitis syndrome Exclusive breastfeeding	Infants <6 mo Formula-fed infants Infectious colitis Hircshsprung’s disease Coagulopathy Anal fissure
Samples provided *				
Feces	7	12	5	5
Milk	8	11	3	4

CMPA, Cow’s milk protein allergy; d, days; wks, weeks; mo, months. Continuous variables are expressed as means ± standard deviation (SD). * Microbiome and immunological analyses were performed in milk and fecal samples provided by the participants, except for one milk sample from the control group and one fecal sample from the CMPA group that could not be processed for the immunological analysis.

**Table 2 nutrients-12-00634-t002:** Frequency and relative abundance of operational taxonomic units at genus level in fecal samples from participants (*N* = 29) in each group of study: control, colic, cow’s milk protein allergy (CMPA), and proctocolitis.

	Control Group (*n* = 7)		Colic Group (*n* = 12)		CMPA Group (*n* = 5)		Proctocolitis Group (*n* = 5)			
Phylum *Genus*	*n* (%)	Median (IQR)	*n* (%)	Median (IQR)	*n* (%)	Median (IQR)	*n* (%)	Median (IQR)	*p* *	*p* **
Actinobacteria										
*Bifidobacterium*	6 (86)	64.24 (61.91–75.05)	12 (100)	62.10 (39.57–88.63)	4 (80)	23.40 (5.27–51.23)	5 (100)	54.14 (37.49–77.62)	0.335	0.040
*Collinsella*	0 (0)	-	4 (33)	4.00 (2.01–14.06)	2 (40)	3.41 (3.37–3.45)	0 (0)	-	0.178	1.000
Bacteroidetes										
*Bacteroides*	4 (57)	2.78 (0.54–5.52)	8 (67)	0.45 (0.22–4.29)	0 (0)	-	4 (80)	1.27 (0.70–3.61)	0.055	0.811
Firmicutes										
*Enterococcus*	5 (71)	0.97 (0.95–1.37)	8 (67)	0.33 (0.20–1.69)	3 (60)	0.42 (0.27–1.42)	5 (100)	3.28 (2.27–3.29)	0.552	0.300
*Lactobacillus*	5 (71)	1.37 (0.69–1.51)	8 (67)	2.10 (0.41–6.31)	3 (60)	0.38 (0.29–1.88)	2 (40)	16.42 (8.67–24.17)	0.822	0.727
*Streptococcus*	7 (100)	2.00 (0.54–8.79)	11 (92)	1.36 (0.46–2.79)	5 (100)	1.67 (1.41–2.54)	5 (100)	0.99 (0.19–1.17)	1.000	0.395
*Erysipelatoclostridium*	5 (71)	8.39 (1.74–8.78)	4 (33)	0.25 (0.18–1.99)	2 (40)	0.38 (0.34–0.43)	1 (20)	0.25	0.333	0.012
*Veillonella*	5 (71)	0.70 (0.39–3.78)	9 (75)	0.49 (0.34–0.76)	3 (60)	1.59 (0.97–12.42)	5 (100)	0.89 (0.20–1.19)	0.579	0.665
Proteobacteria										
*Escherichia*_*Shigella*	5 (71)	9.56 (0.66–14.73)	8 (67)	1.40 (0.47–4.19)	3 (60)	2.42 (1.28–40.69)	5 (100)	6.56 (0.52–51.11)	0.552	0.871
*Klebsiella*	1 (14)	6.32	4 (33)	6.15 (4.26–6.89)	1 (20)	2.81 (2.27–3.34)	3 (60)	0.37 (0.33–6.93)	0.510	0.884
Unclassified_*Enterobacteriaceae*	3 (43)	0.95 (0.85–27.89)	5 (42)	1.21 (0.38–4.62)	4 (80)	4.82 (1.27–19.49)	3 (60)	0.55 (0.35–0.76)	0.579	0.488

CMPA, Cow’s milk protein allergy. Frequency is expressed as the number (%) of samples in which the genus was detected. The relative abundance is expressed as the median (IQR). * Freeman–Halton extension of the Fisher exact probability test for a 2 × 4 contingency table was used to evaluate differences in detection frequencies of the analyzed bacterial genera between groups. ** Kruskal–Wallis tests were used to evaluate differences in the relative abundances of each genus between groups.

**Table 3 nutrients-12-00634-t003:** Frequency and relative abundance of operational taxonomic units at genus level in milk samples from participants (*N* = 26) in each group of study: control, colic, cow’s milk protein allergy (CMPA), and proctocolitis.

	Control Group (*n* = 8)		Colic Group (*n* = 11)		CMPA Group (*n* = 3)		Proctocolitis Group (*n* = 4)			
Phylum *Genus*	*n* (%)	Median (IQR)	*n* (%)	Median (IQR)	*n* (%)	Median (IQR)	*n* (%)	Median (IQR)	*p**	*p* **
Actinobacteria										
*Corynebacterium*	7 (88)	0.88 (0.31–1.76)	10 (91)	0.27 (0.19–0.86)	2 (67)	1.42 (0.78–2.05)	2 (50)	0.51 (0.38–0.64)	0.238	0.735
*Kocuria*	2 (25)	1.18 (1.08–1.29)	4 (36)	0.55 (0.22–6.97)	2 (67)	0.98 (0.77–1.18)	2 (50)	1.38 (0.75–2.01)	0.702	0.865
Firmicutes										
*Bacillus*	4 (57)	2.78 (0.54–5.52)	8 (67)	0.45 (0.22–4.29)	0 (0)	-	2 (50)	3.60 (2.09–5.11)	0.178	0.555
*Gemella*	5 (63)	0.52 (0.31–0.83)	6 (55)	1.89 (0.66–4.71)	1 (33)	0.11	2 (50)	1.75 (1.43–2.07))	0.940	0.150
*Lactobacillus*	5 (63)	0.47 (0.30–1.93)	10 (91)	0.98 (0.39–1.59)	2 (67)	0.86 (0.67–1.04)	3 (75)	1.19 (0.74–2.49)	0.174	0.990
*Staphylococcus*	8 (100)	9.63º (1.88–51.35)	11 (100)	1.93 (1.06–8.61)	2 (67)	6.26 (5.19–7.33)	4 (100)	9.44 (3.24–17.64)	0.115	0.652
*Streptococcus*	8 (100)	17.18 (5.08–48.80)	11 (100)	23.70 (12.72–36.92)	2 (67)	20.48 (16.30–24.66)	4 (100)	35.09 (12.29–65.44)	0.115	0.905
Proteobacteria										
*Acinetobacter*	7 (88)	2.95 (0.59–5.05)	9 (82)	1.32 (0.36–1.52)	3 (100)	0.43 (0.28–0.56)	4 (100)	2.28 (0.22–4.53)	1.000	0.367
*Pseudomonas*	4 (50)	0.37 (0.28–17.02)	6 (55)	0.28 (0.18–0.39)	2 (67)	0.37 (0.34–0.40)	3 (75)	0.96 (0.83–1.11)	0.886	0.303
Unclassified_*Enterobacteriaceae*	8 (100)	9.64 (4.52–19.69)	10 (91)	3.89 (2.82–13.42)	3 (100)	62.68 (49.53–78.66)	3 (75)	1.39 (1.36–31.67)	0.560	0.117

CMPA, Cow’s milk protein allergy. Frequency is expressed as the number (%) of samples in which the genus was detected. The relative abundance is expressed as the median (IQR). * Freeman–Halton extension of the Fisher exact probability test for a 2 × 4 contingency table was used to evaluate differences in detection frequencies of the analyzed bacterial genera between groups. ** Kruskal–Wallis tests were used to evaluate differences in the relative abundances of each genus between groups.

**Table 4 nutrients-12-00634-t004:** Frequency and concentration of immune factors in fecal samples (*N* = 28) from participants in each group of study: control, colic, cow’s milk protein allergy (CMPA), and proctocolitis.

	Control Group (*n* = 7)		Colic Group (*n* = 12)		CMPA Group (*n* = 5)		Proctocolitis Group (*n* = 5)			
Immune factor	*n* (%)	Median (IQR)	*n* (%)	Median (IQR)	*n* (%)	Median (IQR)	*n* (%)	Median (IQR)	*p**	*p***
Innate immunity										
IL1β (ng/L)	6 (86)	533.20 (33.88–1528.44)	11 (92)	33.29 (13.25–112.96)	3 (60)	12.90 (9.30–102.99)	4 (100)	23.80 (13.85–43.85)	0.300	0.593
IL6 (ng/L)	6 (86)	10.75 (6.68–13.93)	12 (100)	9.32 (5.50–11.08)	5 (100)	5.50 (1.00–32.30)	4 (100)	5.50 (4.38–8.13)	0.571	0.796
IL12(p70) (ng/L)	7 (100)	60.30 (48.24–64.85)	12 (100)	37.35 (27.55–48.93)	5 (100)	32.70 (14.20–74.00)	4 (100)	46.45 (22.35–69.40)	1.000	0.492
IFNγ (ng/L)	2 (29)	66.60 (50.50–82.70)	0 (0)		1 (20)	381.70	0 (0)		0.150	0.221
TNFα (ng/L)	6 (86)	48.80 (36.28–89.71)	8 (67)	38.85 (19.01–81.20)	1 (20)	48.80	3 (75)	48.80 (48.80–81.55)	0.152	0.774
Acquired immunity										
IL2 (ng/L)	7 (100)	54.80 (33.20–62.36)	12 (100)	36.80 (20.41–41.30)	5 (100)	51.20 (44.00–51.20)	4 (100)	58.40 (58.40–60.20)	1.000	0.104
IL4 (ng/L)	1 (14)	9.00	0 (0)		0 (0)		0 (0)		0.571	0.392
IL10 (ng/L)	5 (71)	34.60 (27.10–42.20)	10 (83)	37.15 (7.72–89.73)	3 (60)	937.42 (472.71–1105.06)	3 (75)	8.00 (5.80–11.50)	0.811	0.256
IL13 (ng/L)	4 (57)	11.68 (5.90–18.63)	4 (33)	5.10 (3.23–8.58)	0 (0)		2 (50)	15.25 (9.33–21.18)	0.200	0.621
IL17 (ng/L)	7 (100)	66.20 (46.15–120.01)	12 (100)	61.28 (36.10–72.58)	5 (100)	51.10 (51.10–96.80)	4 (100)	26.20 (15.10–48.73)	1.000	0.514
Chemokines										
IL8 (ng/L)	5 (71)	14.30 (9.09–27.00)	7 (58)	4.80 (3.00–7.95)	1 (20)	4.80	3 (75)	11.10 (7.95–17.45)	0.351	0.407
MCP1 (ng/L)	3 (43)	32.50 (28.59–43.15)	2 (17)	4.54 (2.64–6.45)	0 (0)		1 (25)	7.00	0.377	0.145
MIP1β (ng/L)	6 (86)	22.30 (17.58–29.43)	8 (67)	39.19 (16.54–76.90)	1 (20)	381.60	2 (50)	108.70 (100.55–116.85)	0.134	0.279
Hematopoyetic factors										
IL5 (ng/L)	7 (100)	91.90 (86.33–166.85)	12 (100)	59.95 (32.54–110.85)	5 (100)	99.90 (84.00–99.90)	4 (100)	151.15 (131.40–164.88)	1.000	0.301
IL7 (ng/L)	5 (71)	86.30 (74.00–100.37)	5 (42)	8.60 (8.60–35.90)	1 (20)	35.86	1 (25)	98.40	0.352	0.152
GCSF (µg/L)	7 (100)	400.50 (109.60–575.50)	12 (100)	295.15 (119.29–796.05)	3 (60)	445.90 (378.00–869.51)	3 (75)	87.60 (44.60–396.50)	0.036	0.564
GMCSF (µg/L)	6 (86)	12.95 (8.55–18.99)	8 (67)	12.20 (3.63–18.60)	3 (60)	45.30 (27.30–54.13)	3 (75)	15.10 (12.95–17.95)	0.824	0.404

CMPA, Cow’s milk protein allergy; GCSF, granulocyte colony-stimulating factor; GMCSF, granulocyte–macrophage colony-stimulating factor; IFNγ, interferon-γ; IL, interleukin; MCP1, macrophage–monocyte chemoattractant protein-1; MIP1β, macrophage inflammatory protein-1β; TNFα, tumor necrosis factor-α. *n* (%), number (percentage) of samples in which the immunological compound was detected. Concentrations are expressed as median (IQR). * Freeman–Halton extension of the Fisher exact probability test for a 2×4 contingency table was used to evaluate differences in expression frequencies of the analyzed parameters between groups. ** Kruskal–Wallis tests were used to evaluate differences in the concentration of the different immune factors between groups.

**Table 5 nutrients-12-00634-t005:** Frequency and concentration of immune factors in milk samples (*N* = 25) from participants in each group of study: control, colic, cow’s milk protein allergy (CMPA), and proctocolitis.

	Control Group (*n* = 7)		Colic Group (*n* = 12)		CMPA Group (*n* = 5)		Proctocolitis Group (*n* = 5)			
Immune factor	*n* (%)	Median (IQR)	*n* (%)	Median (IQR)	*n* (%)	Median (IQR)	*n* (%)	Median (IQR)	*p* *	*p* **
Innate immunity										
IL1β (ng/L)	6 (86)	0.73 (0.41–1.18)	6 (55)	0.78 (0.19–9.95)	2 (67)	0.22 (0.14–0.29)	4 (100)	3.85 (2.02–8.28)	0.339	0.075
IL6 (ng/L)	4 (57)	5.49 (3.66–9.07)	5 (45)	2.21 (2.21–8.25)	2 (67)	8.94 (5.58–12.31)	4 (100)	2.65 (1.40–4.30)	0.579	0.677
IL12(p70) (ng/L)	4 (57)	4.18 (3.50–5.39)	5 (45)	5.07 (3.28–7.68)	2 (67)	7.89 (7.57–8.21)	3 (75)	2.82 (2.11–4.17)	0.875	0.711
IFNγ (ng/L)	1 (14)	18.24	0 (0)		0 (0)		0 (0)		0.586	-
TNFα (ng/L)	2 (29)	5.14 (3.79–6.48)	4 (36)	3.55 (2.70–5.14)	2 (67)	6.24 (5.98–6.51)	2 (50)	2.44 (1.89–3.00)	1.000	0.594
Acquired immunity										
IL2 (ng/L)	0 (0)		0 (0)		2 (67)	5.69 (4.33–7.06)	0 (0)		0.049	0.002
IL4 (ng/L)	3 (43)	0.33 (0.33–0.75)	1 (9)	0.14	2 (67)	0.69 (0.61–0.76)	3 (75)	0.24 (0.14–0.29)	0.115	0.060
IL10 (ng/L)	4 (57)	2.51 (2.38–4.24)	3 (27)	2.25 (2.12–10.12)	3 (100)	24.00 (13.51–29.46)	1 (25)	0.92	0.371	0.026
IL13 (ng/L)	6 (86)	1.42 (1.07–2.08)a	4 (36)	0.62 (0.31–0.87)b	3 (100)	1.79 (1.08–1.79)ac	4 (100)	2.21 (1.45–3.09)ac	0.126	0.003
IL17 (ng/L)	1 (14)	6.81	0 (0)		2 (67)	20.19 (14.15–26.23)	0 (0)		0.081	0.012
Chemokines										
IL8 (ng/L)	7 (100)	73.19 (25.12–129.25)	11 (100)	39.95 (24.46–45.51)	2 (67)	49.39 (44.76–54.01)	4 (100)	142.55 (113.37–177.55)	0.036	0.091
MCP1 (ng/L)	6 (86)	79.61 (30.43–145.76)	8 (73)	150.45 (86.26–375.50)	3 (100)	157.92 (112.39–307.83)	3 (75)	39.77 (22.96–46.93)	0.688	0.259
MIP1β (ng/L)	7 (100)	14.88 (9.31–44.36)	11 (100)	11.96 (8.69–33.27)	3 (100)	15.53 (9.87–28.88)	4 (100)	26.20 (18.91–38.90)	0.199	0.798
Hematopoyetic factors										
IL5 (ng/L)	0 (0)		0 (0)		0 (0)		1 (25)	1.42	0.345	0.154
IL7 (ng/L)	2 (29)	5.04 (3.17–6.90)	2 (18)	13.87 (7.84–19.89)	1 (33)	15.54	1 (25)	25.24	0.919	0.949
GCSF (µg/L)	3 (43)	17.43 (9.90–17.43)	0 (0)		3 (100)	6.39 (3.56–10.88)	2 (50)	16.35 (14.80–17.90)	0.012	0.017
GMCSF (µg/L)	3 (43)	35.74 (29.11–48.91)	3 (27)	47.33 (42.44–52.07)	2 (67)	582.01 (479.61–684.41)	2 (50)	96.66 (53.01–140.30)	0.779	0.337

CMPA, Cow’s milk protein allergy; GCSF, granulocyte colony-stimulating factor; GMCSF, granulocyte–macrophage colony-stimulating factor; IFNγ, interferon-γ; IL, interleukin; MCP1, macrophage–monocyte chemoattractant protein-1; MIP1β, macrophage inflammatory protein-1β; TNFα, tumor necrosis factor-α. *n* (%), number (percentage) of samples in which the immunological compound was detected. Concentrations are expressed as median (IQR). * Freeman–Halton extension of the Fisher exact probability test for a 2×4 contingency table was used to evaluate differences in expression frequencies of the analyzed parameters between groups. ** Kruskal–Wallis tests were used to evaluate differences in the concentration of the different immune factors between groups.

**Table 6 nutrients-12-00634-t006:** Frequencies of detection of immunological compounds in milk (*n* = 25) and fecal (*n* = 28) samples in each group of study: control, colic, cow’s milk protein allergy (CMPA), and proctocolitis.

Immunological Compound	Control Group			Colic Group			CMPA Group			Proctocolitis Group		
	Milk (*n*/*N*)	Feces (*n*/*N*)	*p* *	Milk (*n*/*N*)	Feces (*n*/*N*)	*p* *	Milk (*n*/*N*)	Feces (*n*/*N*)	*p* *	Milk (*n*/*N*)	Feces (*n*/*N*)	*p* *
Innate immunity												
IL1β	6/7	6/7	1.000	6/11	11/12	0.069	2/3	3/5	1.000	4/4	4/4	1.000
IL6	4/7	6/7	0.559	5/11	12/12	0.005	2/3	5/5	0.375	4/4	4/4	1.000
IL12	4/7	7/7	0.192	5/11	12/12	0.005	2/3	5/5	0.375	3/4	4/4	1.000
IFNγ	1/7	2/7	1.000	0/11	0/12	1.000	0/3	1/5	1.000	0/4	1/4	0.464
TNFα	2/7	6/7	0.102	4/11	8/12	0.220	2/3	1/5	0.464	2/4	3/4	1.000
Acquired immunity												
IL2	0/7	7/7	0.001	0/11	12/12	<0.001	2/3	5/5	0.375	0/4	4/4	0.029
IL4	3/7	1/7	0.559	1/11	0/12	0.478	2/3	0/5	0.107	3/4	0/4	0.143
IL10	4/7	5/7	1.000	3/11	10/12	0.012	3/3	3/5	0.464	1/4	4/4	0.143
IL13	6/7	4/7	0.559	4/11	4/12	1.000	3/3	0/5	0.018	4/4	3/4	1.000
IL17	1/7	7/7	0.004	0/11	12/12	<0.001	2/3	5/5	0.375	0/4	4/4	0.029
Chemokines												
IL8	7/7	5/7	0.462	11/11	7/12	0.037	2/3	1/5	0.464	4/4	4/4	1.000
MCP1	6/7	3/7	0.266	8/11	2/12	0.012	3/3	0/5	0.018	3/4	1/4	0.486
MIP1β	7/7	6/7	1.000	11/11	8/12	0.093	3/3	1/5	0.143	4/4	2/4	0.429
Hematopoyetic factors												
IL5	0/7	7/7	0.001	0/11	12/12	<0.001	0/3	5/5	0.018	1/4	4/4	0.143
IL7	2/7	5/7	0.286	2/11	5/12	0.371	1/3	1/5	1.000	1/4	1/4	1.000
GCSF	3/7	7/7	0.070	0/11	12/12	<0.001	3/3	3/5	0.464	2/4	3/4	1.000
GMCSF	3/7	6/7	0.266	3/11	8/12	0.100	2/3	3/5	1.000	2/4	3/4	1.000

GCSF, granulocyte colony-stimulating factor; GMCSF, granulocyte–macrophage colony-stimulating factor; IFNγ, interferon-γ; IL, interleukin; MCP1, macrophage–monocyte chemoattractant protein-1; MIP1β, macrophage inflammatory protein-1β; TNFα, tumor necrosis factor-α. n/N, number of samples in which the immunological compound was detected/total number of samples assayed. * Fisher exact tests were used to evaluate differences in detection frequencies of the analyzed parameters.

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
