# Peer review of "Microbiological and Immunological Markers in Milk and Infant Feces for Common Gastrointestinal Disorders: A Pilot Study"

_nutrients, 2020, doi:10.3390/nu12030634_

Round 1
Reviewer 1 Report
This manuscript studied the microbiota and immune response from IBD infant feces and maternal milk. This study is an interesting research topic and organized in good shape. I suggest this study with minor spell check.
Author Response
Comments and Suggestions for Authors
This manuscript studied the microbiota and immune response from IBD infant feces and maternal milk. This study is an interesting research topic and organized in good shape. I suggest this study with minor spell check.
Answer: We thank the reviewer for the time you dedicated to our manuscript and for your comment. The manuscript has been checked for spell check, and changes have been made according to the suggestion.

Reviewer 2 Report
Would like to congratulate the author for doing so much work, it is a well-written manuscript with plenty of information. however, there are a few comments from -
Line 55 and 56: citation used here is appropriate and do not indicate that Cows milk allergy is the most prevalent, and author has mentioned that “allergy in childhood in the first year of life” which doesn’t sound right, as childhood is defined as the periods from the age of 2 years to 12 years, whereas period from birth to 1 year is defined as infants and birth to 2 years is defined as infancy. Would suggest the author check the citation and use appropriate terminology.
Did author optimise the kits used for measuring immune markers in human milk samples, as extracting biomarkers from milk samples and getting the middle layers is a tricky task and must have required long laboratory hours? How did author and her team decided to use these particular brands of kits?
In the results section- what are the black dots in Figure 1 b and c? Its is unclear, so describe it in the legends.
Discussion:
Compared to the rest of the manuscript, discussion requires more strengthening. There is a disconnect between discussion and hypothesis or aims. At several places, only results are presented, and it has been left to readers to speculate the implication or significance of these findings and how it may inform future recommendations. Such as line 349 – 350- infants with proctocolitis had reduced IL 10… but what does this information imply and why it is important information.
CMPA group were found different but no statistical difference was found, was it due to small sample size? Did the author check if the mode of delivery was major reason was causing the difference, CMAP rather than CMAP itself, as CMPA group had more C section delivery compared to any other group?
Having a conclusion paragraph could be helpful for the author to support the discussion
Author Response
Would like to congratulate the author for doing so much work, it is a well-written manuscript with plenty of information. however, there are a few comments from -
- Line 55 and 56: citation used here is appropriate and do not indicate that Cows milk allergy is the most prevalent, and author has mentioned that “allergy in childhood in the first year of life” which doesn’t sound right, as childhood is defined as the periods from the age of 2 years to 12 years, whereas period from birth to 1 year is defined as infants and birth to 2 years is defined as infancy. Would suggest the author check the citation and use appropriate terminology.
Answer: We agree with the reviewer 's comment and, therefore, the sentence has been modified as follows:
"Cow’s milk protein allergy (CMPA) is the most prevalent type of food allergy in infancy, and its incidence during first year of life has been estimated to be 2% to 7.5% [new8] [8]."
As a consequence, a new reference related to the incidence of CMPA during the first year of life has been included in the revised manuscript:
Mousan, G.; Kamat, D. Cow’s milk protein allergy. Clin. Pediatr. (Phila) 2016, 55, 1054–1063. doi:10.1177/0009922816664512.
- Did author optimise the kits used for measuring immune markers in human milk samples, as extracting biomarkers from milk samples and getting the middle layers is a tricky task and must have required long laboratory hours? How did author and her team decided to use these particular brands of kits?
Answer: Our laboratory has a long experience in measuring immune markers in human milk samples, including international studies involving several groups worldwide (see doi: 10.3389/fimmu.2017.00696). For this purpose, a big effort was made in optimizing extraction protocols and kits. In all our studies, standard curves are routinely performed for each immune compound. Regarding sample preparation, it requires removal of the fat and particulate suspended material, which is accomplished by a centrifugation step. The protocol for preparation of human colostrum and milk samples was established some years ago, and, since then, it has been used by other research groups working in the same field. It was described for the first time in: Espinosa-Martos, I.; Montilla, A.; de Segura, A.G.; Escuder, D.; Bustos, G.; Pallás, C.; Rodríguez, J.M.; Corzo, N.; Fernández, L. Bacteriological, biochemical, and immunological modifications in human colostrum after Holder pasteurisation. J. Pediatr. Gastroenterol. Nutr. 2013, 56, 560-568. doi: 10.1097/MPG.0b013e31828393ed.
Examples of previous publications where the same protocol was used to measure immune compounds, both in human milk and infant fecal samples, include the following ones:
Moles, L.; Manzano S.; Fernández L.; Montilla A.; Corzo N.; Ares S.; Rodríguez JM.; Espinosa-Martos I. Bacteriological, biochemical, and immunological properties of colostrum and mature milk from mothers of extremely preterm infants. J. Pediatr. Gastroenterol Nutr. 2015, 60,120-6. doi: 10.1097/MPG.0000000000000560.
Espinosa-Martos, I.; Jiménez, E.; de Andrés, J.; Rodríguez-Alcalá, L.M.; Tavárez, S.; Manzano, S.; Fernández, L.; Alonso, E.; Fontecha, J.; Rodríguez, J.M. Milk and blood biomarkers associated to the clinical efficacy of a probiotic for the treatment of infectious mastitis. Benef. Microbes. 2016, 7, 305-18. doi: 10.3920/BM2015.0134.
Ruiz, L.; Espinosa-Martos, I.; García-Carral, C.; Manzano, S.; McGuire, M.K.; Meehan, C.L.; McGuire, M.A.; Williams, J.E.; Foster, J.; Sellen, D.W.; Kamau-Mbuthia, E.W.; Kamundia, E.W.; Mbugua, S.; Moore, S.E.; Kvist, L.J.; Otoo, G.E.; Lackey, K.A.; Flores, K.; Pareja, R.G.; Bode, L.; Rodríguez, J.M. What's normal? Immune profiling of human milk from healthy women living in different geographical and socioeconomic settings. Front. Immunol. 2017, 8, 696. doi: 10.3389/fimmu.2017.00696.
Gomez, M.; Moles, L.; Espinosa-Martos, I.; Bustos, G.; de Vos, W.M.; Fernandez, L.; Rodriguez, J.M.; Fuentes, S.; Jimenez, E. Bacteriological and immunological profiling of meconium and fecal samples from preterm infants: a two-year follow-up study. Nutrients 2017, 9, 1293. doi: 10.3390/nu9121293
Escuder-Vieco, D.; Espinosa-Martos, I.; Rodríguez, J.M.; Fernández, L.; Pallás-Alonso, C.R. Effect of HTST and Holder Pasteurization on the concentration of immunoglobulins, growth factors, and hormones in donor human milk. Front. Immunol. 2018, 9, 2222. doi: 10.3389/fimmu.2018.02222.
Díaz, M.; Guadamuro, L.; Espinosa-Martos, I.; Mancabelli, L.; Jiménez, S.; Molinos-Norniella, C.; Pérez-Solis, D.; Milani, C.; Rodríguez, J.M.; Ventura, M.; Bousoño, C.; Gueimonde, M.; Margolles, A.; Díaz, J.J.; Delgado, S. Microbiota and derived parameters in fecal samples of infants with non-IgE cow's milk protein allergy under a restricted diet. Nutrients. 2018, 10, E1481. doi: 10.3390/nu10101481.
Gómez, M.; Moles, L.; Espinosa-Martos, I.; Bustos, G.; de Vos, W.M.; Fernández, L.; Rodríguez, J.M.; Fuentes, S.; Jiménez, E. Bacteriological and immunological profiling of meconium and fecal samples from preterm infants: a two-year follow-up study. Nutrients 2017, 9, E1293. doi: 10.3390/nu9121293.
De Andrés, J.; Manzano, S.; García, C.; Rodríguez, J.M.; Espinosa-Martos, I.; Jiménez, E. Modulatory effect of three probiotic strains on infants' gut microbial composition and immunological parameters on a placebo-controlled, double-blind, randomised study. Benef Microbes. 2018, 9, 573-584. doi: 10.3920/BM2017.0132.
Finally, we decided to use Mulpiplex bead-based Immunoassays (BioPlex, BioRad) because it is a highly standardized technique that allows to assay a high number or immunological analytes simultaneously from a small amount of a biological sample. This last feature is very relevant when working with human milk since large amounts of samples are not usually available. In addition, we used BioRad as the supplier because of the high quality of kits, reagents, equipment and technical support.
- In the results section- what are the black dots in Figure 1 b and c? Its is unclear, so describe it in the legends.
Answer: The black dots in the box and whisker plots are outliers, i.e. samples that have a value higher (or lower) than 1.5 times the IQR. The explanation has been included in the figure legends.
- Discussion:
Compared to the rest of the manuscript, discussion requires more strengthening. There is a disconnect between discussion and hypothesis or aims. At several places, only results are presented, and it has been left to readers to speculate the implication or significance of these findings and how it may inform future recommendations. Such as line 349 – 350- infants with proctocolitis had reduced IL 10… but what does this information imply and why it is important information.
Answer: We agree that discussion required more strengthening. We have revised this section and included or modified two paragraphs. The first, dealing with differences potentially due to the high number of infants born by C-section in the CMPA group, an observation that was also pointed by the reviewer in his/her next comment (see below). The second in relation to the relevance of the reduced IL10 in infant fecal samples.
P 13, lines 323-326: "This finding might be related to the type of delivery. Most of the infants in CMPA group were born by Cesarean section, which has been related to lower Bifidobacterium content due, at least in part, to delayed initiation of breastfeeding [28]."
P 14, lines 365-367: "IL10 suppresses both the innate and adaptative responses of the immune system, and limits the inflammatory responses [45]. Thus, reduced fecal IL10 could explain bleeding related to gut inflammation in the Proctocolitis group."
- 5. CMPA group were found different but no statistical difference was found, was it due to small sample size? Did the author check if the mode of delivery was major reason was causing the difference, CMAP rather than CMAP itself, as CMPA group had more C section delivery compared to any other group?
Answer: Sometimes, when the sample size is small and there is high variability in the value of the variable, it is possible to find a difference (statistically significant) when all median values are compared (Kruskal-Wallis test) but not in post hoc analysis when medians are compared pairwise because of the correction for multiple comparisons. We did not check if the mode of delivery determined differences between CMAP and other groups, but definitively it could have it, so this comment will be included in the discussion (lines 323-326). See also answer to point 4.
- Having a conclusion paragraph could be helpful for the author to support the discussion
Answer: A conclusion paragraph has been added to the discussion (Lines 391-400).
Reviewer 3 Report
Aparicio et al. presented pilot comparisons of microbiological and immunological markers in milk and infant feces and for common gastrointestinal disorders, and found some differences in bacterial families and immune factors. It is of interest to the research community, however, there are major issues that need to be addressed.
Major comments
The authors appeared to stretch the results to find significance, and some of the conclusions appeared overstated. Along these lines, some of the statistical tests seemed questionable, or need justification.
In line 195, paired comparisons between groups were made, even though comparisons among groups were not significant. And in the Figure 1 legend, Montecarlo tests were used to compare differences between groups. Montecarlo tests were not mentioned in the Methods. And the conventional approach is to do comparisons among groups first, using ANOVA or Kruskal Wallis test, and if significant, post hoc comparisons are made between groups, using Tukey's test or equivalent non-parametric test, with correction for multiple comparisons. Why was Montecarlo test used directly for paired comparisons between groups? What the authors did and presented is confusing and needs more explanation/justification. This is also the case for Figure 3.
For statistical tests, the chi-square test was used to compare frequencies between two groups, but in Table 6, the majority of the tests were Fisher's Exact tests. Why did the authors choose one over another test in these cases? Most of the tests have similar numbers of N, and where possible, the Fisher's Exact test should be performed instead of the chi-square test. Similar situation in the Table S1.
Since the conclusions are not remarkably strong/robust, especially when the sample size is small, some of the language should be toned down a bit to reflect this. For example, in line 32, '...may serve to differentiate...' can be changed to something like 'a few microbial signatures in feces may explain part of the difference between CMPA and other infants.'
Minor comments
Line 57, are mostly involved -> plays a major role
Lines 78-79, were suffering from any of the gastrointestinal disorders described above.
Line 106, each extracted DNA sample
Lines 124-126, After quality control, ... by using QIIME package... (did the authors meant TM as a superscript? Usually QIIME with citation is norm)
Line 169, most abundant genera
Line 342, reveal notable changes in either the frequency of detection or the concentration of the
Line 372, should be tested in further studies...
Author Response
Major comments
- The authors appeared to stretch the results to find significance, and some of the conclusions appeared overstated. Along these lines, some of the statistical tests seemed questionable, or need justification.
In line 195, paired comparisons between groups were made, even though comparisons among groups were not significant. And in the Figure 1 legend, Montecarlo tests were used to compare differences between groups. Montecarlo tests were not mentioned in the Methods. And the conventional approach is to do comparisons among groups first, using ANOVA or Kruskal Wallis test, and if significant, post hoc comparisons are made between groups, using Tukey's test or equivalent non-parametric test, with correction for multiple comparisons. Why was Montecarlo test used directly for paired comparisons between groups? What the authors did and presented is confusing and needs more explanation/justification. This is also the case for Figure 3.
Answer: We apologize for several mistakes in section 3.1. Metataxonomic profiling of the fecal samples.
First, in line 169, “general” should be replaced by “bacterial phyla”. In addition, no specific data about phyla were included in Table 2. Therefore, the sentence would be: “No differences were observed regarding to the frequency of detection or concentration of the most abundant bacterial phyla between groups (results not shown)”.
Second, Table 2 shows the relative abundance of OTUs at the genus level (the more abundant and frequent genera). There were two errors in the transcription of data when construction Table 2. The p values resulting from the Kruskal-Wallis test to compare the median values of the relative abundance of Bifidobacterium and Erysipelatoclostridium should be 0.040 and 0.012, respectively.
Third, in Figure 3 we intended to show the differences found when the post hoc analysis (Dunn test) was performed after the Kruskal-Wallis test in order to know which pair of groups were different. The Montecarlo test in the legend of Figures 1 and 3 is a cut and paste error.
In addition, the post hoc comparisons for pairs of samples (Dunn tests) were performed at the taxonomical levels where the Kruskal-Wallis tests indicated that there was a statistically significant difference. However, this happened only in bacterial families and genera that were present in low amounts in the samples (and, therefore, they were not included in the tables), except for Bifidobacterium and Erysipelatoclostridium in fecal samples.
The section 2.6. Statistical and bioinformatic analysis has been changed to improve the clarity of the analysis performed.
- For statistical tests, the chi-square test was used to compare frequencies between two groups, but in Table 6, the majority of the tests were Fisher's Exact tests. Why did the authors choose one over another test in these cases? Most of the tests have similar numbers of N, and where possible, the Fisher's Exact test should be performed instead of the chi-square test. Similar situation in the Table S1.
Answer: We usually apply the Fisher’s exact test when any of the expected frequencies per null hypothesis is lower than 5. However, taking into account the small sample size of our groups, we believe that it would be more appropriate to use the Fisher’s exact test, as the reviewer suggests. Therefore, we performed this analysis for all the tables in the manuscript and supplementary material. However, in most of the comparisons there were no differences in the p values calculated with both tests. When the comparison was between four groups, the Freeman-Halton extension of the Fisher exact probability test for a 2×4 contingency table was used.
- Since the conclusions are not remarkably strong/robust, especially when the sample size is small, some of the language should be toned down a bit to reflect this. For example, in line 32, '...may serve to differentiate...' can be changed to something like 'a few microbial signatures in feces may explain part of the difference between CMPA and other infants.'
Answer: We agree with the reviewer that we may have emphasized the results having into account the number of participants. Therefore, we have changed the conclusion in the abstract and in the discussion:
Lines 32-33: “In conclusion, a few microbial signatures in feces may explain part of the difference between CMPA and other infants.”
Lines 386-388: “However, we identified a few microbial signatures in feces that may explain part of the difference between CMPA and other infants. We also detected some milk immunological signatures among the different conditions addressed in this study.”
Minor comments
Answer: We are grateful for these corrections because they have improved the quality of the manuscript.
Line 57, are mostly involved -> plays a major role
Answer: Corrected. See lines 57-58: “Proctocolitis accounts for up to 64% of rectal bleeding in the infant, and CMPA or lactose intolerance plays a major role in its development [9].”
Lines 78-79, were suffering from any of the gastrointestinal disorders described above.
Answer: Corrected. See lines 77-80: “In this context, the objective of this work was, first, to assess the microbiome and a wide spectrum of immunological parameters in maternal milk and infant feces from groups of mother-infant pairs in which the infants were suffering from any of the different gastrointestinal disorders described above.”
Line 106, each extracted DNA sample
Answer: Corrected. See lines 105-108: “Purity and concentration of each extracted DNA sample was estimated using a NanoDrop 1000 spectrophotometer (NanoDrop Technologies, Inc., Rockland, USA).”
Lines 124-126, After quality control, ... by using QIIME package... (did the authors meant TM as a superscript? Usually QIIME with citation is norm)
Answer: Corrected. See lines 126-127: “After a quality control, the reads were assembled and the resulting sequences were processed by using QIIME package version 1.9.1 [19,20]”
Line 169, most abundant genera
Answer: Corrected according to the explanation given in point 1 of this document. See lines 170-171: “No differences were observed regarding to the frequency of detection or concentration of the most abundant bacterial phyla between groups (results not shown).”
Line 342, reveal notable changes in either the frequency of detection or the concentration of the
Answer: Corrected. See lines 356-358: “Our study did not reveal notorious changes in either the frequency of detection or the concentration of the immunological factors in feces,…”
Line 372, should be tested in further studies...
Answer: Corrected. See lines 388-389: “Their usefulness as biomarkers should be tested in further studies involving a much larger population.”